# Cell-autonomous immune gene expression is repressed in pulmonary neuroendocrine cells and small cell lung cancer

Ling Cai[1,2,3✉], Hongyu Liu[1,4], Fang Huang[2], Junya Fujimoto[5], Luc Girard[3,6,7], Jun Chen[4,8], Yongwen Li [4], Yu-An Zhang[6], Dhruba Deb[6], Victor Stastny[6], Karine Pozo[9], Christin S. Kuo[10], Gaoxiang Jia[1], Chendong Yang[2], Wei Zou[11], Adeeb Alomar[6], Kenneth Huffman[6], Mahboubeh Papari-Zareei[6], Lin Yang[12], Benjamin Drapkin[3,6,9], Esra A. Akbay [13], David S. Shames[11], Ignacio I. Wistuba[5], Tao Wang [1,3], Jane E. Johnson [3,7,14], Guanghua Xiao[1,3,15], Ralph J. DeBerardinis [2,3,16], John D. Minna [3,6,7,9,18✉], Yang Xie[1,3,15,18✉] & Adi F. Gazdar[3,6,13,17,18]

Small cell lung cancer (SCLC) is classified as a high-grade neuroendocrine (NE) tumor, but a subset of SCLC has been termed "variant" due to the loss of NE characteristics. In this study, we computed NE scores for patient-derived SCLC cell lines and xenografts, as well as human tumors. We aligned NE properties with transcription factor-defined molecular subtypes. Then we investigated the different immune phenotypes associated with high and low NE scores. We found repression of immune response genes as a shared feature between classic SCLC and pulmonary neuroendocrine cells of the healthy lung. With loss of NE fate, variant SCLC tumors regain cell-autonomous immune gene expression and exhibit higher tumor-immune interactions. Pan-cancer analysis revealed this NE lineage-specific immune phenotype in other cancers. Additionally, we observed MHC I re-expression in SCLC upon development of chemoresistance. These findings may help guide the design of treatment regimens in SCLC.

[1] Quantitative Biomedical Research Center, Department of Population and Data Sciences, UT Southwestern Medical Center, Dallas, TX, USA. [2] Children's Research Institute, UT Southwestern Medical Center, Dallas, TX, USA. [3] Simmons Comprehensive Cancer Center, UT Southwestern Medical Center, Dallas, TX, USA. [4] Tianjin Key Laboratory of Lung Cancer Metastasis and Tumor Microenvironment, Tianjin Lung Cancer Institute, Tianjin Medical University General Hospital, Tianjin, China. [5] Department of Translational Molecular Pathology, University of Texas MD Anderson Cancer Center, Houston, TX, USA. [6] Hamon Center for Therapeutic Oncology Research, UT Southwestern Medical Center, Dallas, TX, USA. [7] Department of Pharmacology, UT Southwestern Medical Center, Dallas, TX, USA. [8] Department of Lung Cancer Surgery, Tianjin Lung Cancer Institute, Tianjin Medical University General Hospital, Tianjin, China. [9] Department of Internal Medicine, UT Southwestern Medical Center, Dallas, TX, USA. [10] Department of Pediatrics, Stanford University, Stanford, CA, USA. [11] Department of Oncology Biomarker Development, Genentech Inc., South San Francisco, CA, USA. [12] Department of Pathology, National Center/Cancer Hospital, Chinese Academy of Medical Sciences and Peking Union Medical College, Beijing, China. [13] Department of Pathology, UT Southwestern Medical Center, Dallas, TX, USA. [14] Department of Neuroscience, UT Southwestern Medical Center, Dallas, TX, USA. [15] Department of Bioinformatics, UT Southwestern Medical Center, Dallas, TX, USA. [16] Howard Hughes Medical Institute, UT Southwestern Medical Center, Dallas, TX, USA. [17]Deceased: Adi F. Gazdar. [18]These authors jointly supervised this work: John D. Minna, Yang Xie, Adi F. Gazdar. ✉email: Ling.Cai@UTSouthwestern.edu; John.Minna@UTSouthwestern.edu; Yang.Xie@UTSouthwestern.edu

Small cell lung cancer (SCLC), accounting for 15% of lung cancer cases, with a 5-year survival of 6%, is designated by the US Congress as a "recalcitrant cancer"[1,2]. SCLC is classified as a high-grade neuroendocrine (NE) tumor[3]. A large fraction of SCLC tumors are driven by *ASCL1*, a lineage oncogene also important for pulmonary neuroendocrine cell (PNEC) fate determination[4,5]. In healthy lung tissue, PNECs are rare and dormant[6], whereas upon lung injury, some act as stem cells to regenerate surrounding epithelial cells[7]. SCLC occurs primarily in heavy smokers, but despite the very high mutation burden[8–10] from SCLC genomes predicted to contribute an ample supply of neoantigens, SCLCs express low levels of major histocompatibility complex class I (MHC I) proteins to present tumor-specific antigens[11,12]. This could explain why, among various types of cancer, checkpoint-blockade immunotherapy underperforms in SCLC[13,14].

Thirty-five years ago, it was observed that by contrast to classic SCLC cell lines (which grew in tissue culture as floating cell aggregates), a subset of patient-derived SCLC lines behaved differently—growing as adherent monolayers in culture, with morphologically larger cells, more prominent nucleoli, and expressed few or no NE markers[15,16]. These characteristics led such tumors to be termed "variant" or "non-NE" SCLC. Many of these variant SCLC lines were established from patients whose tumors had acquired resistance to chemotherapy and clinically relapsed, a context in which genomic *MYC* amplification was also noted to be more frequent[17]. Notch activation had been shown to mediate the transition from classic to variant subtypes and accounts for the intratumoral heterogeneity commonly seen in SCLC[18].

Recently, extending the concepts of classic and variant SCLC, both intertumoral, and intratumoral heterogeneity in SCLC has been documented and has been associated with the expression of lineage-specific transcription factors (TFs) *ASCL1*, *NEUROD1*, *YAP1*, and *POU2F3*, and these various subtypes express different levels of NE markers[19–21].

We have previously defined a 50-gene NE signature that helps us quantify the NE properties as a continuous NE score ranging from −1 to 1, with a more positive score indicating higher NE properties[22]. In the current study, we applied this NE scoring method to SCLC samples from preclinical models and patient tumors. We first assessed the relationship between NE scores and SCLC molecular subtypes. Then, we investigated the immune phenotypes associated with variable NE scores in SCLC and other cancer types.

## Results

### Relationship between NE scores and SCLC molecular subtypes.

Using the 50-gene NE signature updated with all available SCLC-related RNA-seq data (Supplementary Data 1), we computed NE scores for patient-derived SCLC lines and xenografts (PDXs) as well as four independent patient tumor data sets (including one newly generated for this study) (Table 1 and Supplementary Data 2). We examined the relationship between NE scores and expression of SCLC molecular subtype-specific TFs as proposed by Rudin et al. (Fig. 1a, b). Our findings are largely consistent with the previous proposal that assigns ASCL1[+] and NEUROD1[+] SCLCs to NE subtypes and POU2F3[+] and YAP1[+] SCLCs to non-NE subtypes. However, we note some discrepancies. First, we found that while expression of *ASCL1* and *NEUROD1* seems to be mutually exclusive in cell lines, they seem to co-express in many of the tumor samples; a small set of samples with low NE scores still express *ASCL1* or *NEUROD1*; in "George_2015", "Jiang_2016" and our own data set, we have observed rare POU2F3[+] samples that have high NE scores.

With serially sectioned formalin-fixed paraffin-embedded (FFPE) slides from 9 out of the 18 tumors for which we had performed expression profiling, we examined the tumors with hematoxylin and eosin (H&E) staining as well as immunohistochemistry (IHC) staining of ASCL1, NEUROD1, and POU2F3 (Fig. 1c–f). The high NE-score tumors exhibited predominantly classic SCLC morphology with dark nuclei, scant cytoplasm, and inconspicuous nucleoli. Notably, this was not only seen in ASCL1[+] tumors (for example, SCLC-04, NE score 0.4) but also in the POU2F3[+] tumor with a positive NE-score (SCLC-15, NE score 0.26) (Fig. 1c). On the other hand, while we observed variant morphology in tumors with low NE scores, we noticed intratumoral heterogeneity. In a tumor weakly positive for ASCL1 (SCLC-20, NE score -0.05), the ASCL1-high regions were found to be more classic-like whereas the ASCL1-low regions were more variant-like (Fig. 1d). Our IHC-based quantifications largely agree with the microarray gene expression assessments (Fig. 1e). Tumors that were found to express both *ASCL1* and *NEUROD1* stained positive for both markers as well. In addition, intratumoral heterogeneity was commonly found within such tumors, which contain areas with high expression of both TFs but also areas with expression of only one TF (Fig. 1f).

### Immune gene repression is a NE lineage-specific property.

We performed a correlation between NE scores and SCLC transcriptomic data to identify gene expression changes associated with the NE program. Not surprisingly, gene ontology (GO) analyses revealed genes related to the neuronal system as highly expressed in high NE-score samples (Supplementary Fig. 1a, b). By contrast, genes negatively associated with NE score were enriched for GO terms related to immune response, and this was observed in both the cell line and human tumor data sets (Supplementary Fig. 1c, d). We also performed gene set enrichment analysis (GSEA)[23] with a variety of gene set libraries collected by Enrichr[24]. Consistent with the previous report that Notch signaling dependent REST (Neuron-Restrictive Silencer Factor) activation represses neuronal gene expression in variant SCLC[18], we found REST targets (i.e., repressed by REST) are abundantly expressed in high NE-score SCLCs. On the other hand, interferon-stimulated genes (ISGs) are found to highly expressed in the low NE-score (variant) SCLC samples (Fig. 2a, b and Supplementary Data 3). As NFκB signaling mediates activation of ISGs, we examined reverse-phase protein array (RPPA) data from the cancer cell line encyclopedia (CCLE)[25] and found higher levels of activating serine 536 phosphorylation on p65[26] in low NE-score SCLC lines (Fig. 2c). This result aligns with the previous findings that innate immune genes are more highly expressed in an SCLC cell line H69 variant cell line with upregulation of mesenchymal genes—it was found that in this cell line, TBK1, IRF3, and STAT1 signaling is activated by a subclass of endogenous retroviruses (ERVs) through MAVS and STING to form a positive feedback loop and sustain innate immune gene expression[27]. As classic SCLC is driven by lineage factor *ASCL1* whereas variant SCLC with loss of NE genes is driven by *YAP1*, we examine the functional roles of *YAP1* and *ASCL1* in regulating the ISGs in SCLC cell lines. Horie et al. previously examined the consequences of silencing *YAP1* in YAP1[+] SCLC cell lines[28]. From their results, we observed a significant decrease of ISG expression in SCLC cell line SBC5 without upregulation of NE genes targeted by REST. ASCL1 silencing from an ASCL1+ SCLC cell line H2107 on the other hand[29], did not increase the expression of ISGs (Supplementary Fig. 2).

Our 50-gene signature derived from lung cancer cell line mRNA data (Supplementary Data 1) contains several genes with immune-related functions that were found to highly express in

**Table 1 Data sets used for analyses.**

| Source | Name | Tissue source | Sample type | n | References |
|---|---|---|---|---|---|
| Human | SCLC cell lines/ NCI/Hamon Center | SCLC | Cell line | 69 | This study |
| Human | Horie_2016 | SCLC | Cell line | 4 | 28 |
| Human | Pozo_2020 | SCLC | Cell line | 4 | 29 |
| Human | Cañadas_2014 | SCLC | Cell line | 6 | 56 |
| Human | Drapkin_2018 | SCLC | PDX | 19 | 57 |
| Human | Rudin_2012 | SCLC | Tumor | 29 | 9 |
| Human | George_2015 | SCLC | Tumor | 81 | 10 |
| Human | Jiang_2016 | SCLC | Tumor | 79 | 80 |
| Human | SCLC tumors (this study) | SCLC | Tumor | 18 | This study |
| Human | expO | Lung cancer | Tumor | 109 | 82 |
| Human | Rousseaux_2013 | Lung cancer | Tumor | 286 | 81,82 |
| Human | CCLE | Pan-cancer | Cell line | _b | 25 |
| Human | TCGA | Pan-cancer | Tumor | 10535 | 53 |
| Human | TARGET | Pan-cancer | Tumor | 734 | 53 |
| Mouse | Lim_2017 | SCLC | Pooled FACS-sorted tumor cells[a] | 6 | 18 |
| Human | Travaglini_2020 | Healthy lung | Single cell | 9384[c] | 42 |
| Mouse | Ouadah_2019 | Healthy lung | Single cell | 46[c] | 7 |

*PDX* patient-derived xenografts, *FACS* fluorescence-activated cell sorting, *scRNA-seq* single-cell RNA sequencing.
[a]In Lim_2017, Rb1$^{flox/flox}$;p53$^{flox/flox}$;p130$^{flox/flox}$;Rosa26$^{mTmG}$; Hes1$^{GFP/+}$ GEMM SCLC tumors were initiated by Ad-CMV-Cre, sorted by Tomato and GFP to obtain relatively pure tumor cells.
[b]CCLE data sets were used in multiple analyses with different numbers of cell lines; [c]Cells.

variant SCLC. Some are involved in cytokine signaling; for example, *IL18* encodes for a proinflammatory cytokine[30], and *OSMR* encodes for a receptor for oncostatin M and IL-31[31]. Furthermore, many of these genes are involved with immuno-suppressive processes, including *NT5E*[32], *TGFBR2*[33], *ANXA1*[34], *EPHA2*[35], *HFE*[36], and *LGALS3*[37]. From our pathway analysis results, NE score negatively correlated genes are also enriched for genes upregulated in response to Tumor Necrosis Factor-alpha (TNFA) or interleukin-1 (IL1) (Supplementary Data 3). indicates that beyond these genes included in the NE expression signature, there is a broad immune program concertedly upregulated in low NE-score SCLC samples. We extended our analysis to a few immune gene sets that were previously identified to express cell-autonomously in cancer. These gene sets include the following: SPARCS genes (stimulated 3 prime antisense retroviral coding sequences) reported to express in mesenchymal tumors and mediate interferon-gamma signal amplification[27]; "parainflammation" genes in epithelial tumor cells[38]; and senescence-associated secretory phenotype (SASP) genes[39] that reinforce the senescence arrest, alter the microenvironment, and trigger immune surveillance of the senescent cells[40]. We observed that the expression of these genes also negatively correlate with NE scores in SCLC despite little overlap among genes in these various sets (Supplementary Fig. 2a, b). We also assessed about 1,000 innate immune genes cataloged by the InnateDB[41] and found a significantly higher proportion of these genes negatively associated with NE score from SCLC cell lines (Supplementary Fig. 3c and Supplementary Data 4).

While the expression of neuronal program genes in high NE-score SCLCs can be attributed to the NE lineage, we examined single-cell RNA-seq (scRNA-seq) data from the healthy human lung epithelial cells[42] to check whether the expression repression of ISGs is also a lineage-specific phenomenon that could be observed in PNECs rather than being cancer-specific. Consistent with the previous report that ASCL1 negatively regulates YAP1 during neuronal differentiation[43], the highest expression of *ASCL1* and lowest expression of *YAP1* was observed in PNECs, relative to other cell types. We confirmed that while PNECs have increased expression of REST target genes, ISGs are indeed repressed as well (Fig. 2d and Supplementary Fig. 4a). Additionally, we specifically

examined interferon receptors in PNECs and found that they also have the lowest expression in PNECs (Fig. 2d). It has been estimated that 10% of the genes in the human genome have the potential to be regulated by IFN, many ISGs work in immune defense against viral infection, but some could be hijacked by viruses[44]. As some PNECs are rare stem cells, we reason that ISG repression might lower their risk from viral infection. In the context of the current COVID-19 pandemic, we examined scRNA-seq data from Ouadah et al., who performed lineage tracing with an *Ascl1*$^{CreERT2}$; *Rosa26*$^{LSL-ZsGreen}$ mouse model to show that some PNECs can transdifferentiate into other cell types[7]. Supplementary Fig. 4b generated with their data shows that AT2 and ciliated cells originated from PNECs in this model have lost *Ascl1* but increased *Yap1* expression. *Ly6e* and *Tmprss2*, genes involved in coronavirus defense[45] and hijacked entry[46], respectively, were also upregulated.

**Increased tumor–immune interaction in low NE-score SCLC tumor samples.** It has been long observed that the expression of MHC I is low in SCLC[11]. Using single sample gene set enrichment analysis (ssGSEA)[47], we derived the MHC I scores for MHC I genes. From studies that had collected lung tumors of different histology, MHC I scores positively correlate with *PTPRC* (which encodes for pan-leukocyte marker CD45) levels (Fig. 3a and Supplementary Fig. 5a). The lowest MHC I and *PTPRC* gene expression were found in neuroendocrine tumors, including not only SCLC but also carcinoids (Fig. 3a), suggesting these NE tumors with decreased MHC I have fewer immune infiltrates. In SCLC data sets, low NE-score samples exhibited upregulation of MHC I genes (Fig. 3b) and were associated with higher *PTPRC* expression in patient tumor data sets (Supplementary Fig. 5b). We also estimated immune cell infiltration by deriving immune cell type-specific signature scores[48] and found that they negatively correlate with NE scores in SCLC patient tumors, suggesting increased tumor–immune interaction in low NE-score tumors (Fig. 3c).

We saw a higher expression of PD-L1 (CD274) in low NE-score SCLCs for 3 out of 4 primary tumor data sets (Supplementary Fig. 6a). Furthermore, genes from an IFN-gamma related signature that has been shown to predict PD-1

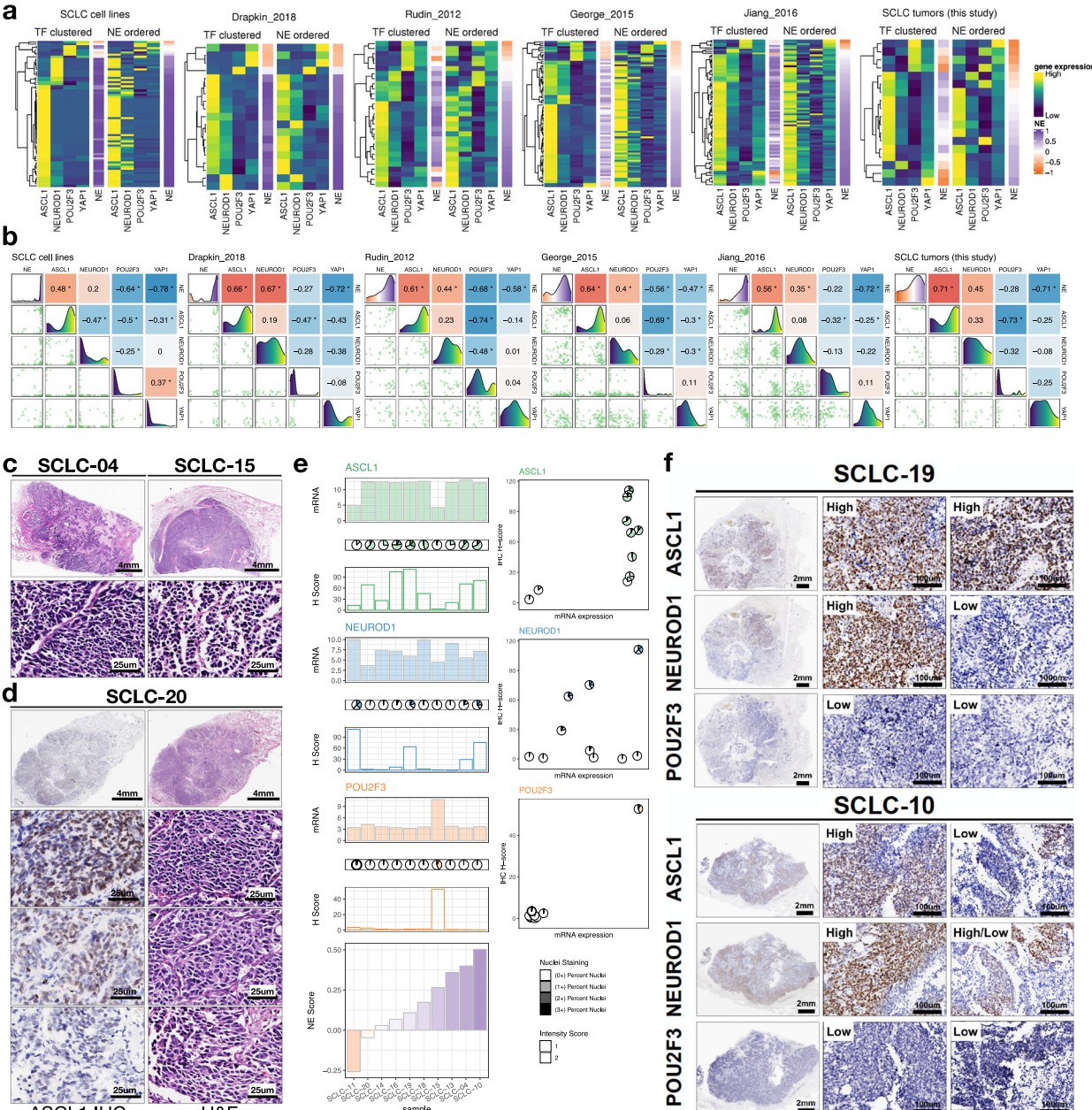

**Fig. 1 NE score and SCLC molecular subtypes. a** Heatmaps visualizing relative expression of molecular subtype-specific TFs and NE scores. Two heatmaps were generated for each study, with one ordered by complete linkage hierarchical clustering of TFs and the other ordered by NE scores. Gene expression was z-score standardized for each sample. **b** Pairs plot for NE scores and molecular subtype-specific TFs. Lower left panels are scatter plots showing a pairwise relationship between variables, diagonal panels are density plots showing the distribution of each variable and upper right panels are correlation coefficients from pairwise Pearson correlation. Refer to the top or right of each 5 × 5 matrix for subplot x or y axis variable labels. For example, for each data set, the first subplot on the top row shows the distribution of NE scores from that data set, the scatterplot below it shows the relationship between NE score (x axis value) and *ASCL1* expression (y axis value), and the Pearson correlation coefficient between NE score and *ASCL1* expression is provided in the second cell of the top row. *p-value < 0.05. Note that in all studies: most samples have positive NE score; under TF-based classification, ASCL1+ subtype dominates; NE scores positively correlate with *ASCL1* and *NEUROD1* but negatively correlate with *POU2F3* and *YAP1* expression. **c** H&E staining of two high NE-score SCLC tumor samples showing classic SCLC morphology with dark nuclei, scant cytoplasm, and inconspicuous nucleoli. **d** ASCL1 IHC staining and H&E staining of a low NE-score SCLC tumor, showing variable morphology at different selected areas, where ASCL1-low areas appear to be more variant-like. **e** Quantifications of TF expression from IHC staining or microarray profiling, samples are ordered by increasing NE scores. **f** IHC of ASCL1, NEUROD1, and POU2F3 in two tumors that express both ASCL1 and NEUROD1. Two areas per tumor were selected for showing intratumoral heterogeneity in ASCL1 and NEUROD1 expression patterns.

blockade response in multiple cancer types[49] are highly expressed in low NE-score SCLC tumors across multiple data sets (Fig. 3c). We also examined a list of 21 immune checkpoint genes[50], immune-suppressive cytokines (IL-10 and TGF-beta), and their receptors[51], for their association with NE scores. We found that these genes also have higher expression in low NE-score SCLC tumors (Fig. 3c). Finally, the expression of 995 immunosuppressive genes from the Human Immunosuppression Gene Atlas[50]

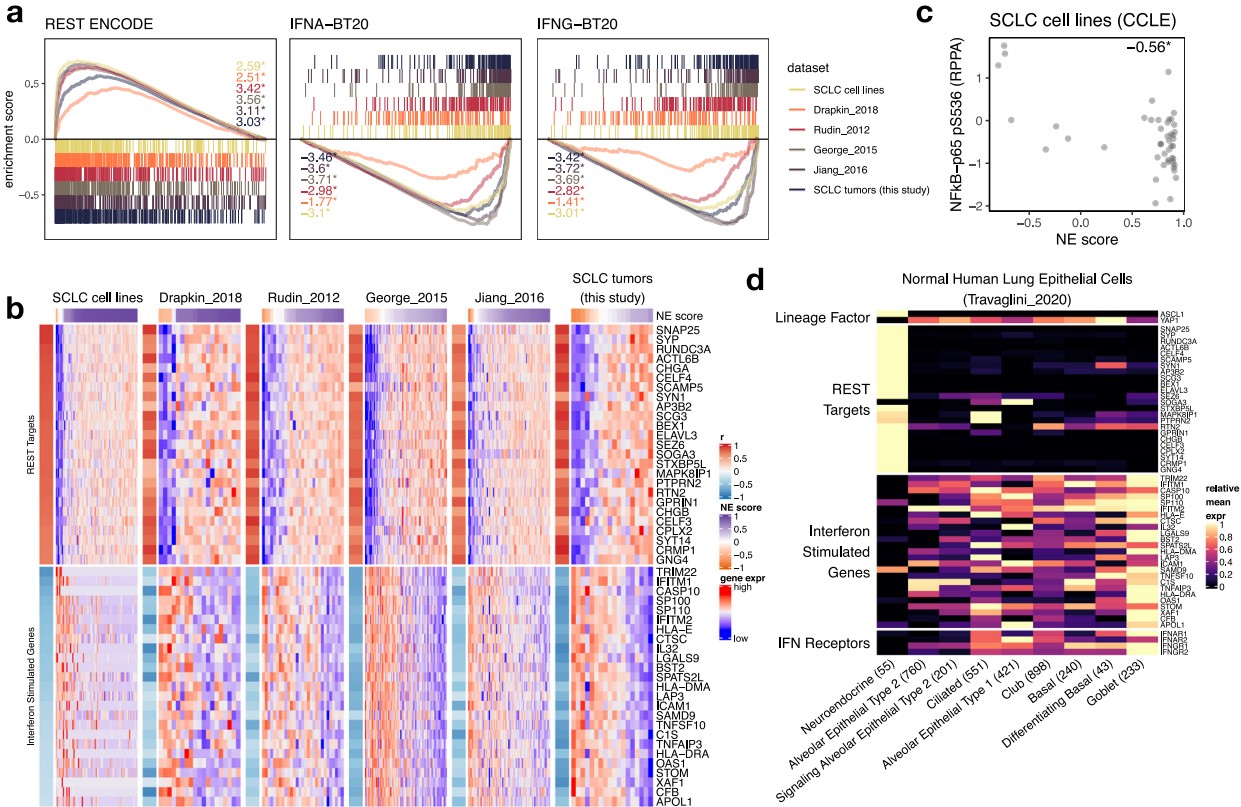

**Fig. 2 Repression of ISGs in high NE-score SCLC and PNECs. a** GSEA enrichment plots for selected gene sets. Results from SCLC cell lines, PDXs (Drapkin_2018), and patient tumor data sets were superimposed. Normalized enrichment score (NES) was provided. *, multiple comparison-adjusted *p*-value < 0.05. **b** Heatmaps for top 25 leading-edge genes selected from gene sets in **a**. Gene expression matrix of each data set was annotated with color-coded Pearson correlation coefficient (from correlating NE score with gene expression) as a left-side column, and a top bar indicating NE scores. For each data set, gene expression was *z*-score standardized across samples. **c** Scatter plot showing negative correlation between NE score and Ser536 phosphorylation on NFkB-p65 in SCLC cell lines. Pearson correlation coefficient was provided. **p*-value < 0.05. **d** Heatmap showing relative expression of selected lineage factors (*ASCL1* and *YAP1*), REST targets, and ISGs (same genes as used in **b**, determined from **a**) as well as interferon receptor genes in healthy human lung epithelial cells based on scRNA-seq experiments. Gene expression from all cells in each cell-type cluster was averaged and then min–max scaled across cell types to a scale between 0 and 1.

was assessed, and again, the majority of these genes exhibit a negative correlation between mRNA expression and NE scores across different SCLC tumor data sets (Supplementary Fig. 6b and Supplementary Data 5).

Besides gene expression-based analyses, we also performed immunohistochemistry (IHC) with our 9 SCLC tumor samples to quantify tumor-infiltrating CD8+ and CD4+ T cells (Supplementary Data 2). Of importance, both intratumoral and intertumoral heterogeneity were observed in T cell infiltration. Within the same tumor, areas with low tumor ASCL1 levels exhibited more CD8+ and CD4+ T cell infiltration, whereas areas with high tumor ASCL1 levels showed fewer CD8+ or CD4+ T cells (Fig. 4a). Across all the SCLC tumor specimens assessed, CD8+ and CD4+ T cell per area cell count positively correlated with the T cell score computed from gene expression data, and both IHC-based T cell counts and gene expression-based T cell scores negatively correlated with NE scores (Fig. 4b).

**Pan-cancer analyses for NE score expression and immune response genes.** These findings had prompted us to examine other cancer types to see whether immune gene repression is seen in other NE tumors and whether variant subtype from NE lineage loss could also be observed (Fig. 4c). A recent study identified SCLC-like epithelial tumors in pan-cancer samples using a principal component analysis-based approach. They found that tumors across many lineages with a higher SCLC-like score had lower immune gene expression[52]. We applied our NE scoring method across all cancer lineages (pan-cancer analysis) to compute NE scores and assess their relationship with immune phenotypes. In pediatric (Therapeutically Applicable Research to Generate Effective Treatments—TARGET) and adult (The Cancer Genome Atlas—TCGA) pan-cancer studies[53], neuroendocrine tumor neuroblastoma (NBL), as well as pheochromocytoma & paraganglioma (PCPG) were identified as containing the highest NE scores (Fig. 5a, b). Tumors of glial origin, including Low-Grade Glioma (LGG) and Glioblastoma Multiforme (GBM) also have high NE scores. Besides these NE/glial tumors, a small number of high NE-score samples were observed for bladder urothelial carcinoma (BLCA), breast invasive carcinoma (BRCA), lung adenocarcinoma (LUAD), lung squamous cell carcinoma (LUSC), pancreatic adenocarcinoma (PDAC) and stomach adenocarcinoma (STAD), for which it is also known that neuroendocrine tumors, while uncommon, still comprise a small proportion of the cases (Fig. 5a, b). In a previous immunogenomic analysis that had classified pan-cancer TCGA samples into six immune subtypes[54], we found samples from the "immunologically quiet" subclass have the highest NE score, followed by the "lymphocyte depleted" subclass (Fig. 5c). We further assessed the relationship between NE scores and the tumor-infiltrating lymphocytes and leukocyte regional fractions previously reported for the pan-cancer samples[54], these immune metrics negatively correlate with NE scores across all samples (Fig. 5d) and also within specific tumor types (Supplementary Fig. 7).

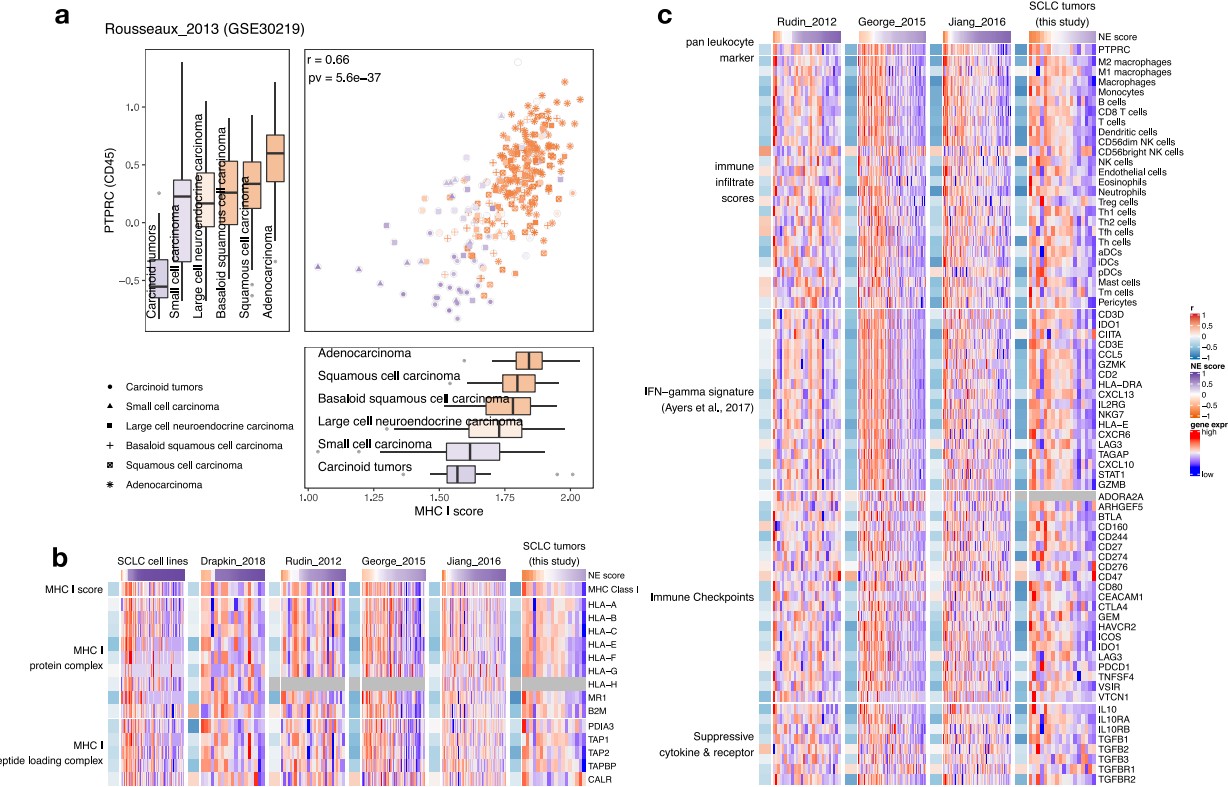

**Fig. 3 Low NE-score variant SCLC has increased tumor–immune interaction. a** Expression of MHC I genes and pan-leukocyte marker *PTPRC* in lung tumors from the "Rousseaux_2013" data set. Box whisker plots are filled with color reflecting the median NE score in different histological subtypes. The centerline in the boxplot represents the median, the lower and upper hinges correspond to the first and third quartiles, the whiskers extend from the hinge to the value no further than 1.5 * IQR away. Color for scatterplot symbols reflects NE score for different samples. **b** Heatmaps visualizing expression of MHC I genes across multiple SCLC data sets. **c** Heatmaps visualizing expression of *PTPRC*, immune-cell-type-specific signature scores, PD-1 blockade response-predicting IFN-gamma-related signature genes[49], immune checkpoint genes, and suppressive cytokines and receptors in SCLC tumor data sets. MHC I score and immune infiltrate scores were computed using ssGSEA methodology. Sample-wise *z*-score standardized values were used for the heatmaps.

We took a close examination of NBL using cell line expression data from CCLE[25] along with patient tumor data from TARGET[55] for lineage factors *ASCL1* and *YAP1*, REST targets, ISGs, MHC I, immune cell-specific signature scores[48], Ayer et al.'s PD-1 blockade response signature[49], immune checkpoints[50], and suppressive cytokines and receptors[51]. The pattern for NBL (Fig. 5e) highly resembles that of SCLC (Figs. 2b and 3b, c) suggesting the existence of a variant NBL subset with decreased neuroendocrine features, increased cell-autonomous expression of immune genes as well as increased tumor–immune interaction. Like SCLC, we also found higher levels of NFkB-p65 phosphorylation in the low NE-score variant NBL cell lines (Fig. 5f).

**MHC I re-expression in chemoresistant SCLC.** As it was previously observed that variant SCLC cell lines were frequently derived from patients whose tumors had relapsed on chemotherapy[17], we wondered if the development of chemoresistance in tumors was associated with the altered expression of immune genes, especially MHC I. Five sets of data with origin-matched chemosensitive and chemoresistant samples were examined to address this question. In 2017, using a genetically engineered mouse model (GEMM), Lim et al. showed that Notch-active SCLC cells were more chemoresistant[18]. Using their data we found the Notch-active SCLC cells had switched from ASCL1+ to YAP1+, have reduced NE scores, and increased expression of ISGs and MHC I genes (Fig. 6a). We next examined a series of preclinical models we and others have developed for human SCLC. Classic,

high NE-score SCLC cell lines predominantly grow as floating aggregates in culture, but contain a small proportion of cells growing adherently in a monolayer. By selecting for adherent growth, we generated an adherent subline H69-AD(/NCI-H69-AD) from parental, chemosensitive H69(/NCI-H69) cells (Fig. 6b). Increased resistance to Cisplatin (~10 fold) and Etoposide (~6 fold) was observed in H69-AD compared to the parental H69 cells (Fig. 6c). We found H69-AD had transitioned to become a low NE-score (-0.02) YAP1+ variant line compared to the parental high NE-score (0.91) classic ASCL1+ line. Both ISGs and MHC I genes were found to have increased expression in H69-AD (Fig. 6d). In a previous study, Cañadas et al. also derived sublines from H69. Hepatocyte growth factor treatment was used to induce the mesenchymal transition of H69 cells, resulting in H69-M lines that were found to be chemoresistant both in vitro and in vivo[56]. From their data set, we also found MHC I expression increased in H69-M compared to parental H69 cells. There was some increase in ISGs too, but less prominent compared to H69-AD from us. Notably, although *YAP1* expression increased in H69-M, *ASCL1* levels did not change (Fig. 6d). As the fourth set of data, PDX models established sequentially from SCLC tumors (Drapkin_2018) collected before and after chemotherapy from the same patient[57] were examined. In PDXs from patient MGH1518 for which chemoresistance had developed in the relapsed sample, we found upregulation of MHC I, but not ISGs (Fig. 6d). Of note, this relapsed sample maintained a high NE score but expressed higher levels of *MYC*, consistent with previous findings that MYC

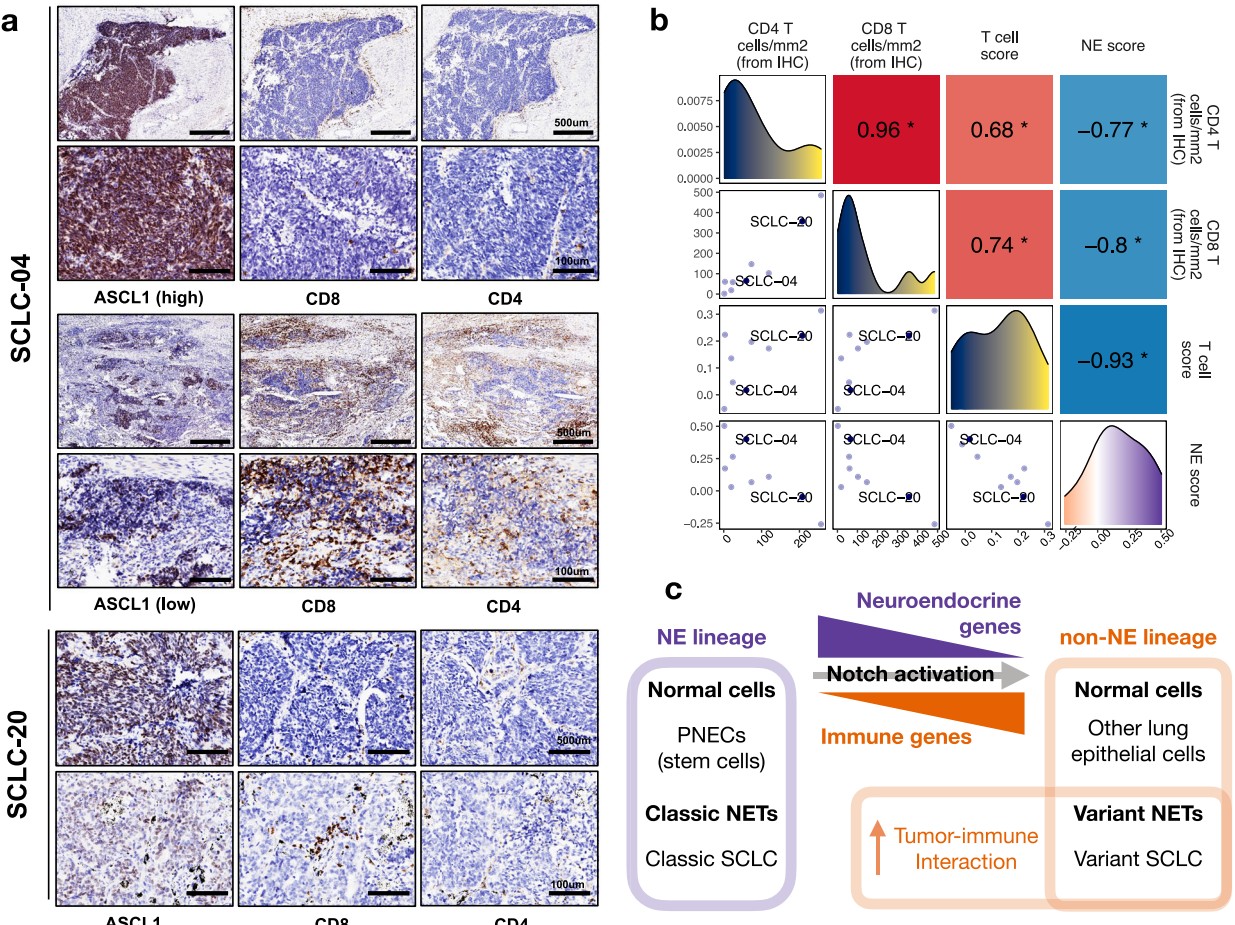

**Fig. 4 Intertumoral and intratumoral heterogeneity in T cell infiltration from SCLC tumors with variable NE features. a** IHC of ASCL1, CD4, and CD8 in selected tumors. SCLC-04 is an SCLC tumor with an NE score of 0.4. CD8 or CD4 T cells were few in the ASCL1-high regions but abundant in the ASCL1-low regions; SCLC-20 is a tumor with NE score of −0.05, similar reciprocal relationship of ASCL1 staining and T cell infiltration was observed. Representative regions with high or low ASCL1 staining were shown. **b** Relationship between IHC-determined per area CD4 and CD8 T cell count, gene expression-based T cell score, and NE score in all nine tumors assessed. Lower left panels are scatter plots showing a pairwise relationship between variables, diagonal panels are density plots showing the distribution of each variable and upper right panels are correlation coefficients from pairwise Pearson correlation. *p-value < 0.05. T cell score was calculated by the ssGSEA method. **c** Schematic diagram showing a relationship between neuroendocrine and immune gene expression in normal cells and neuroendocrine tumors (NETs).

mediates chemoresistance[57]. Lastly, we generated a set of sub-cutaneous xenograft models from a high NE-score human SCLC cell line NCI-H1436 with or without selection for resistance to Cisplatin and Etoposide in mice (Fig. 6e). Compared to the parental xenograft, the drug-resistant xenografts maintained *ASCL1* expression but exhibited increased *B2M* (MHC I complex subunit), *PSMB8* (immunoproteasome subunit) (Fig. 6e), and *MYC*[58]. Collectively, these findings suggest MHC I can re-express upon the development of chemoresistance—in some cases, with lineage transition; and in some other cases, accompanied by an increase in *MYC* expression.

We also checked whether expression levels of MHC I and *MYC* differ by tumor source and anatomical site based on the "NCI/Hamon Center" patient-derived SCLC lines data set (Supplementary Fig. 8). Interestingly, the lowest MHC I and *MYC* levels were both observed for cell lines derived from primary site lung tumor specimens and they are all high-NE tumors, whereas higher MHC I levels were observed in SCLC lines isolated from metastatic tumor samples especially those from lymph node and bone marrow. These observations remain to be validated with primary and metastatic samples from the same patients.

## Discussion

In this study, we examined NE properties of patient-derived SCLC cell lines, PDXs, and human tumors based on NE scores estimated from a gene expression signature. Currently, it is believed that Notch activation drives the lineage transition from ASCL1[+] to NEUROD1[+] to YAP1[+] subtype[59], whereas POU2F3 + SCLC is a standalone subtype that originated from tuft cells[60]. While we observed mutually exclusive patterns of *ASCL1* and *NEUROD1* expression in cell lines, their co-expression was identified in many patient tumors. Our IHC results further revealed intratumoral heterogeneity in such tumors, suggesting ongoing lineage transition in primary treatment-naive tumors. From the alignment of NE scores and molecular subtype-specific TF expression, we observed rare high NE-score POU2F3[+] tumors in three independent data sets, raising the possibility that POU2F3[+] tumors could also arise from NE lineage. Does this represent the fact that all of the respiratory epithelium arises from a common stem cell that then differentiates into sub-niches giving rise to the other differentiated cells? Such concept would also explain the observations in GEMMs[61,62] that SCLC may occasionally arise from other non-NE cells of the respiratory epithelium.

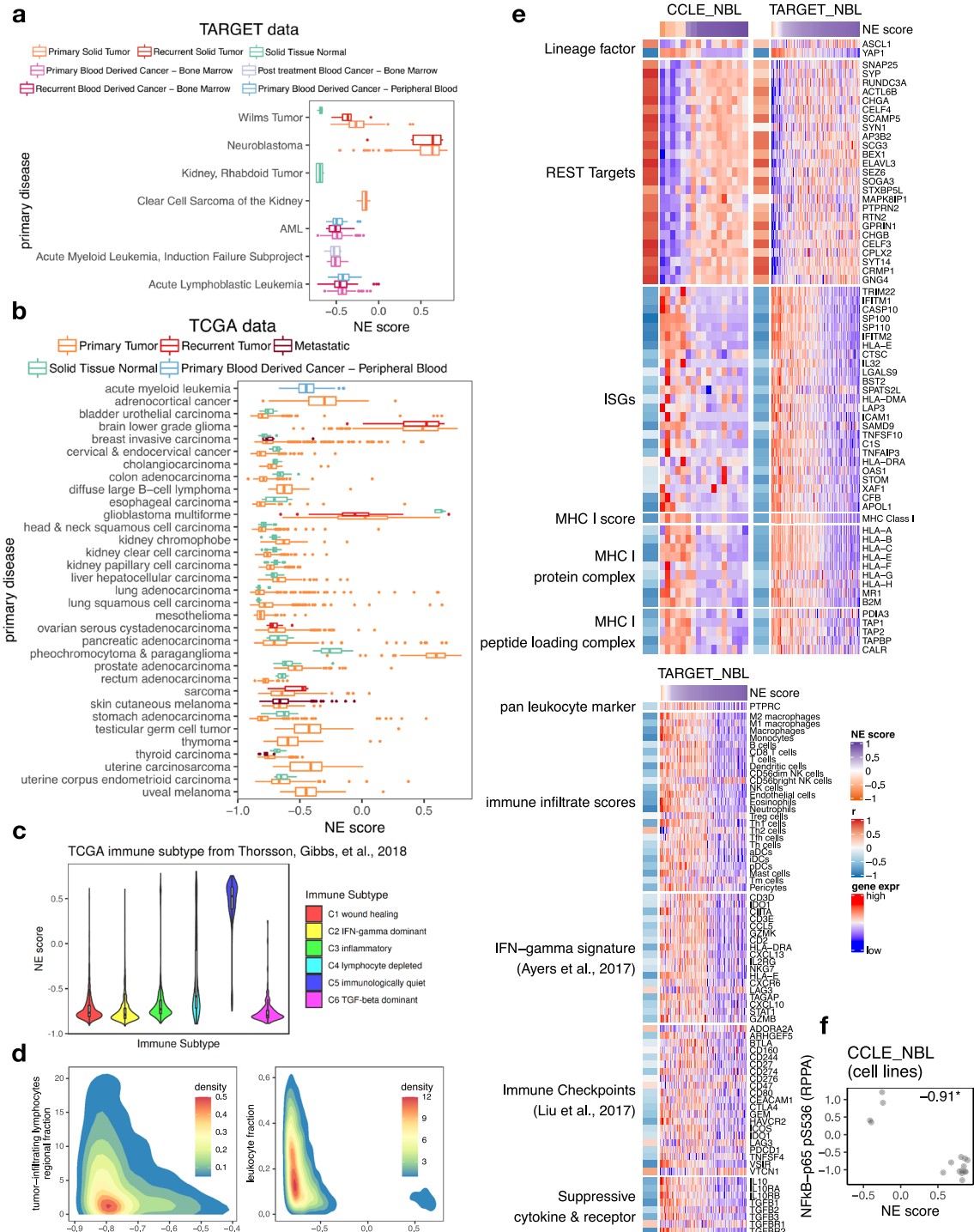

**Fig. 5 Relationship between NE scores and immune phenotypes in pan-cancer samples. a, b** NE scores of pan-cancer samples in the TARGET pediatric cancer cohorts (**a**) and TCGA adult cancer cohorts (**b**). **c** NE scores by immune subtype in TCGA pan-cancer samples. **d** 2D density plots visualizing the relationship between NE scores and tumor-infiltrating lymphocytes regional fraction (Pearson correlation coefficient −0.15, p-value < 2.2e−16) or leukocyte fraction in TCGA pan-cancer samples (Pearson correlation coefficient −0.31, p-value < 2.2e−16). **e** Heatmap visualizing expression of various genes and summary scores previously assessed for SCLC and now in NBL with cell line and tumor data sets side-by-side. For each data set, sample-wise z-score standardized values were used. **f** Scatter plots showing a negative correlation between NE score and Ser536 phosphorylation on NFkB-p65 in NBL cell lines. Pearson correlation coefficient was provided. *p-value < 0.05. For all box whisker plots, the centerline represents the median, the lower, and upper hinges correspond to the first and third quartiles, the whiskers extend from the hinge to the value no further than 1.5 * IQR away.

Our investigation of immune phenotypes associated with variable NE scores had identified repression of ISGs in classic high NE-score SCLC. While it remains to be determined what other pulmonary cells besides PNECs can function as cells of origin for SCLC[61–65], the gene expression similarities between PNECs and SCLC suggests many of the SCLC properties could be tied to PNEC characteristics. We confirmed ISG repression in PNECs relative to other lung epithelial cells through the examination of scRNA-seq data from

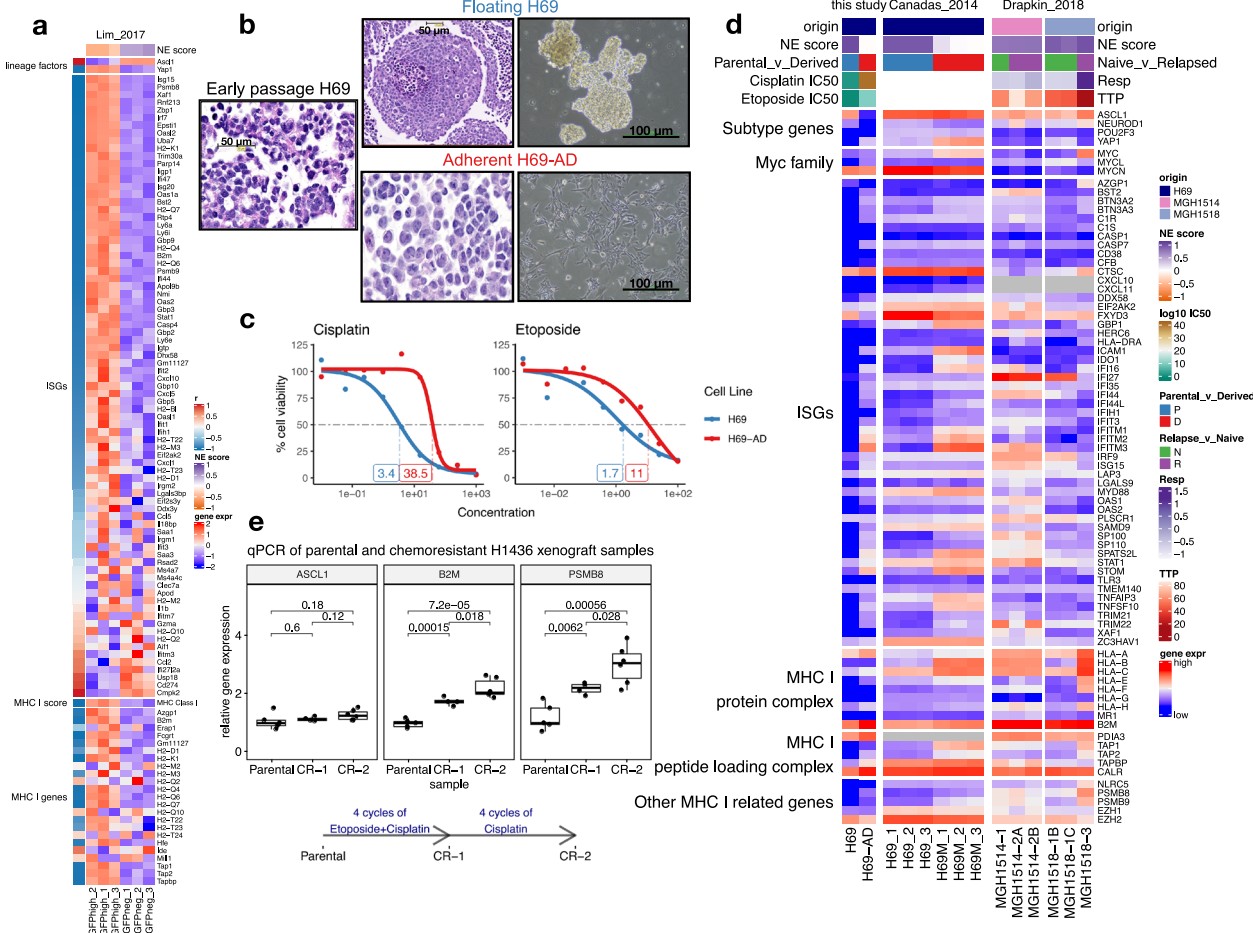

**Fig. 6 MHC I upregulation in chemoresistant SCLC. a** Heatmap visualizing increased expression of ISGs, MHC I genes as cells switch from Ascl1+ to Yap1+ in SCLC GEMM tumors from the "Lim_2017". GFP was expressed from an endogenous promoter of a Notch target gene *Hes1* in *Rb⁻/⁻/p53⁻/⁻/p130⁻/⁻* (TKO) background. Using flow cytometry, the authors first sorted out SCLC tumor cells and then further sorted by GFP to obtain relatively pure tumor cells with different Notch activation status. Three biological replicates were each provided for GFP negative (Notch inactive, classic high-NE) cells and GFP-positive cells (Notch-active, low-NE). **b** Different morphology and culture characteristics of adherent H69-AD and the parental H69. **c** Dose–response curves for Cisplatin and Etoposide in the H69 cell line pair. Note that H69-AD, the adherent line, is more resistant with higher IC50s. **d** Expression changes of selected genes in H69 cell line pair from this study, H69 and derived mesenchymal H69-M cell lines from "Canadas_2014" and autologous PDX samples before and after chemotherapy from "Drapkin_2018". PDX parameters: TTP, time to progression, defined by time to 2× initial tumor volume; RESP, change in tumor volume between initial tumor volume and a minimum of days 14–28. Relapsed sample from MGH-1514 did not show increased chemoresistance based on the RESP and TTP parameters. Note that unlike other heatmaps, due to the small number of samples in each data set, the expression is not scaled across samples in this heatmap. **e** qPCR measurement of normalized *ASCL1*, *B2M*, and *PSMB8* expression in naive parental and chemoresistant H1436 xenograft tumors. *p*-values are based on a two-sided *t*-test. The centerline in the boxplot represents the median, the lower and upper hinges correspond to the first and third quartiles, the whiskers extend from the hinge to the value no further than 1.5 * IQR away.

healthy human lungs. As a limitation of our current study, the biological significance and mechanism of ISG repression in healthy and cancer cells of NE lineage remains to be determined with additional functional studies. ISGs provide viral defense for cells but some can be hijacked by viruses[44]. Since PNECs assume stem cell roles for tissue regeneration after injury[7], lowering expression of ISGs and other genes involved in viral entry presumably have some role in self-protection. While we observed repression of basal ISG expression in PNECs, it remains to be determined in the presence of interferon stimulation whether PNECs would also be less primed for further activation of ISGs. Like SCLC, PNECs can switch from ASCL1+ to YAP1+ through Notch activation, but in the context of tissue repair[7]. Our findings suggest that the increase of cell-autonomous immune gene expression as high NE-score SCLCs transition to low NE-score SCLCs are mirroring the changes that normally take place during the transdifferentiation of PNECs to other lung epithelial cell types, and in SCLC, this had, in turn, led to

increased tumor–immune interaction. Interestingly, our pan-cancer analysis had extended this finding of a reciprocal relationship between neuroendocrine and immune gene expression to other cancer types. For NBL, a cancer with drastically distinct etiology compared to SCLC, we were also able to identify more inflammatory variant cancer cell line and tumors with loss of NE lineage gene expression. It would be interesting to explore more of such molecular similarities between SCLC and other tumors with NE/neuronal lineage.

The full repertoire of immune evasion strategies employed by SCLC remains to be elucidated. However, our results tying together with current clinical treatment findings raise several important questions and paradoxes. The first paradox is that we found a depletion of immune infiltrates in high NE-score SCLC tumors and other neuroendocrine tumors (NETs), associated with the down-regulation of MHC I expression. While this MHC I expression explains the presence of very few T cells, it raises the question of how

high NE-score NETs evade natural killer cells that normally would recognize the missing self that such MHC I expression loss conveys[66]. Thus, we feel the high NE-score low-MHC I expression pairing indicates we need to understand how natural killer cell mechanisms are avoided in NET pathogenesis. The second paradox is that low NE-score variant SCLCs appear to be associated with expression of MHC I and a more immune infiltrated tumor microenvironment, yet clinical trials of immune checkpoint-blockade do not clearly show these are the tumors responding to such therapy. Since we found these tumors also express many immunosuppressive genes it will be important to know which of these immunosuppressive gene functions need to be targeted to achieve anti-tumor immune responses. Finally, we observed the expression of MHC I in selected SCLC samples with chemoresistance and increased *MYC* expression even without changes in ASCL1 expression. We need to know whether immunosuppressive mechanisms are the same or different in the high vs. low NE-score SCLC resistant to chemotherapy. We conclude, that some 30 years after the first description of classic (high NE-score) and variant (low NE-score) SCLCs there are important links between these NE phenotypes and the expression of immune phenotypes, and between similar gene expression profiles of SCLC and pulmonary neuroendocrine cells. Importantly, these correlations identify important problems to be solved of clinical therapeutic translational relevance.

## Methods

**Computation of NE score**. The construction of the original NE signature has been described by Zhang et al.[22]. In this study, this signature has been updated with expression data from RNA-seq experiments. A quantitative NE score can be generated from this signature using the formula: NE score = (correl NE – correl non-NE)/2, where correl NE (or non-NE) is the Pearson correlation between expression of the 50 genes in the test sample and expression of these genes in the NE (or non-NE) cell line group. This score has a range of −1 to +1, where a positive score predicts for NE while a negative score predicts for non-NE cell types. The higher the score in absolute value, the better the prediction.

**Pathway enrichment analysis with GO terms**. Gene Ontology enRIchment anaLysis and visuaLizAtion tool (GOrilla[67], http://cbl-gorilla.cs.technion.ac.il/) was used to identify enriched GO terms[68] related to biological processes (BP) from gene lists ranked by increasing or decreasing Pearson correlation with NE scores in cell line data sets or "George_2015" tumor data set. The *p*-value threshold was set at $10^{-3}$ for resulting GO terms. The output was visualized by Treemap R scripts generated from "reduce + visualize gene ontology" (REViGO[69], http://revigo.irb.hr/) and further customized with a modified color scheme.

**Gene set enrichment analysis (GSEA)**. Gene set libraries were downloaded from Enrichr[24] (https://amp.pharm.mssm.edu/Enrichr/). Fast GSEA based on gene label permutation from R package "fgsea"[70] was first used for a fast screening across a large number of gene set libraries. After reviewing the results for SCLC cell lines, sample label permutation-based GSEA[23] was run for selected gene set libraries to obtain normalized enrichment scores and multiple comparison-adjusted *p*-values. Pearson correlation was used as the ranking metric from correlating gene expression with NE scores.

**Visualization**. All heatmaps were generated by R package "ComplexHeatmap"[71]. Other R packages used for visualization include "ggplot2"[72], "ggridges"[73], "ggrepel"[74], "ggpubr"[75], "treemap"[76], "RColorBrewer"[77], "jcolors"[78] and "patchwork"[79].

**Expression data**. Horie_2016[28] is a set of microarray data that examines the effect of YAP1 KD in SCLC cell lines, the author-processed data were downloaded from GEO with accession id GSE93400. Pozo_2020[29] is a set of RNA-seq data that examines the effect of ASCL1 KD in SCLC cell lines, FPKM data was obtained from the author. The original data have been deposited to GSE151002. Drapkin_2018[57] is a set of RNA-seq data for SCLC PDX. Transcript per million (TPM) data processed by the original authors were downloaded from GEO with accession id GSE110853; Rudin_2012[9] is a set of RNA-seq data for human SCLC tumors, it was obtained from the authors; George_2015[10] is a set of RNA-seq data for human SCLC tumors, FPKM (fragments per kilobase of exon per million fragments mapped) data processed by the original authors were obtained from a supplementary table of the original publication; Jiang_2016 is a set of RNA-seq data for human SCLC tumors[80] DESeq normalized read count data was downloaded from GEO with accession id GSE60052; IGC's Expression Project for Oncology—expO (GSE2109) and Rousseaux_2013 (GSE30219)[81] are microarray data for lung cancer

samples with different histological subtypes, processed previously for the lung cancer explorer (LCE)[82]. Pan-cancer RNA-seq data from TCGA and TARGET processed by Expectation-Maximization (RSEM) algorithm was downloaded from Toil xena hub[53]. CCLE cell line RPPA data, as well as TPM RNA-seq data, were downloaded from the DepMap portal (19Q1)[25]. Travaglini_2020 is a set of scRNA-seq data from healthy human lung[42]. Author-processed count data were downloaded from Synapse with accession id syn21041850. FACS-sorted SmartSeq2 data was used. Cell types with less than 10 cells were removed from analyses. Ouadah_2019 is a set of scRNA-seq data from healthy mouse lung[7]. TPM data were downloaded from GEO with accession id GSE136580. For data from GEO, R package GEOquery[83] was used for extracting the expression and phenotype data. Quantile normalization was performed for bulk expression data by running the "normalize.quantiles" function from R package "preprocessCore"[84]. Log 2 transformation of gene expression data was performed as necessary. Library size normalization was performed for author-processed scRNA-seq data by running the "library.size.normalize" function from R package "phateR"[85].

**Gene signatures**. SPARCS gene set is from a study by Cañadas et al. (Fig. 1S in original article)[27]. Parainflammation gene set is from a study by Aran et al. (Fig. 1C in original article)[38]. SASP gene set is from a study by Ruscetti et al. (Fig. 2C in original article)[86]. InnateDB genes were downloaded from InnateDB[41], non-human genes were filtered out.
    Gene set "REST ENCODE" is from the "ENCODE_and_ChEA_Consensus_TFs_from_ChIP-X" library, and "IFNA-BT2" and "IFNG-BT2" are from the "LINCS_L1000_Ligand_Perturbations_up" library. Both libraries were downloaded from Enrichr[24]. The top 25 genes from the leading edge and are common to all SCLC data sets were selected for heatmap visualization. For ISGs, the leading edge genes from "IFNA-BT2" and "IFNG-BT2" were first combined and then the top 25 genes were selected.
    Mouse ISGs, from Cilloniz et al.[87], were identified from interferome[88] by specifying "mouse" as the species of interest and "lung" as the organ of interest. An unfiltered ISG set was used for Fig. 6A.
    Human MHC I gene set is a combination of genes under GO terms "GO_MHC_CLASS_I_PROTEIN_COMPLEX" and "GO_MHC_CLASS_I_PEPTIDE_LOADING_COMPLEX" from Molecular Signatures Database (MSigDB)[23,89]. Mouse MHC I genes were selected from GO: 0019885, "antigen processing and presentation of endogenous peptide antigen via MHC class I" based on the Mouse Genome Informatics (MGI) database[90]. Immune-cell-specific gene sets in human are from DisHet[48]. Interferon-gamma signature that predicts response to PD-1 blockade is from Ayers et al.[49]. The 21-gene immune checkpoint set and 995-gene immunosuppressive set are from HisgAtlas, a human immunosuppression gene database[50].

**MHC I and Immune infiltrate scores**. R package GSVA[91] was used to compute immune infiltrate scores by single sample GSEA (ssGSEA) method[23,47].

**Patients and tissue specimens**. Study participants included 18 patients who were diagnosed with SCLC and underwent surgical resection of lung cancer between 2006 and 2010 at the Department of Lung Cancer Surgery, Tianjin Medical University General Hospital. Written informed consent was obtained, and the institutional ethics committee of Tianjin Medical University General Hospital approved the study. The cases were selected based on the following criteria: (1) diagnosis of primary lung cancer clinical stage I to IV (pTNM); (2) undergoing surgical resection. Pathologic diagnosis was based on WHO criteria. Lung cancer staging for each patient has performed according to the AJCC Cancer Staging Manual, 8th edition, and was based on findings from physical examination, surgical resection, and computed tomography of the chest, abdomen, pelvis, and brain. The following information was collected from the patients' medical records: age, gender, clinical stage, pathologic diagnosis, differentiation, lymph node status, metastasis, smoking status, and overall survival time. Resected lung and lymph node tissues were immediately immersed in liquid nitrogen until RNA extraction.

**Immunohistochemistry—histology and immunohistochemistry**. Tissue blocks, once collected, were reviewed by a staff thoracic pathologist to confirm SCLC histology. Consecutive four-micrometer-thick tissue sections were cut for immunohistochemistry. IHC staining was performed with a Bond Max automated staining system (Leica Microsystems Inc., Vista, CA) using IHC parameters optimized previously. Antibodies used in this study included ASCL1 (dilution 1:25; Clone 24B72D11.1, BD Biosciences, Cat# 556604), NEUROD1 (dilution 1:100; Clone EPR20766, Abcam, ab213725), POU2F3 (dilution 1:200; polyclonal, Novus Biologicals, NBP1-83966), CD4 (dilution 1:80; Leica Biosystems, CD4-368-L-CE-H) and CD8 (dilution 1:25; Thermo Scientific, MS-457s) in a Leica Bond Max automated stainer (Leica Biosystems Nussloch GmbH). The expression of proteins was detected using the Bond Polymer Refine Detection kit (Leica Biosystems, Cat# DS9800) with diaminobenzidine as chromogen[92]. The slides were counterstained with hematoxylin, dehydrated and cover-slipped. FFPE cell lines pellets with known expression of ASCL1, NEUROD1, and POU2F3 were used to establish and optimize IHC conditions and assess sensitivity and specificity for each antibody.

**Immunohistochemistry—image analysis**. The stained slides were digitally scanned using the Aperio ScanScope Turbo slide scanner (Leica Microsystems Inc.) under ×200 magnification. The images were visualized by ImageScope software (Leica Microsystems, Inc.) and analyzed using the Aperio Image Toolbox (Leica Microsystems Inc.). Different intensity levels of ASCL1, NEUROD1, or POU2F3 nuclear expression were quantified using a 4-value intensity score (0, none; 1, weak; 2, moderate; and 3, strong) and the percentage (0–100%) of the extent of reactivity. A final expression score (H-score) was obtained by multiplying the intensity and reactivity extension values (range, 0–300) as previously described[93]. For example, for SCLC-04, (3+)% nuclei is 1.22, (2+)% nuclei is 9.97, (1+)% nuclei is 47.81, the resulting H-score is $3*1.22 + 2*9.97 + 1*47.81 = 71.41$.

The lymphocyte cells expressing CD4+ and CD8+ were counted by a pathologist using Aperio Image Toolbox analysis software (Aperio, Leica Biosystems) and expressed as cell density (CD4+ and CD8+ cells/mm$^2$ of analyzed tissue)[92,94].

**Microarray assay**. The Human Genome U133 Plus 2.0 microarray with 54,000 probe sets was purchased from the Affymetrix (Lot#: 4032359). Total RNA was extracted with the Trizol reagent (Invitrogen) from the tissue samples. The extracted RNA was purified using the Oligotex mRNA Midi Kit (Qiagen). Then double-strand cDNA synthesis was made using a one-cycle cDNA synthesis kit (Affymetrix) and purified again by column followed by the synthesis of complementary RNA (cRNA) with in vitro transcription (IVT) kit (Affymetrix). The cRNA was fragmented after purification by column and the quality was verified by ultraviolet spectrophotometer and 1.2% denaturing agarose gel. After the test gene-chip (Lot#: 4020852, Affymetrix) was affirmed satisfactory, the real chip hybridization of cRNA fragmentation was performed and then stained and washed. Finally, the real chip was scanned in an Affymetrix scanner and the data were collected by GCOS (gene-chip operation software). CEL files were read into an AffyBatch object by "AffyBatch" function under the "affy"[95] R package. Alternative cdf package[96] "hgu133plus2hsentrezg" was downloaded from "http://mbni.org/customcdf/22.0.0/entrezg.download/hgu133plus2hsentrezg.db_22.0.0.zip" and was specified in the function so that the resulting expression data were processed to gene level rather than the original probe level. (Probe name follows format concatenating Entrez ID for the gene and "_at". For example "3939_at" corresponds to gene LDHA). The AffyBatch object was then converted to an expression set using robust multi-array average (RMA) expression measure by running the "rma" function under R package "affy". Quantile normalization was performed by running "normalize.quantiles" function from the R package "preprocessCore"[84].

**RNA-seq**. RNA samples from SCLC cell lines ($n = 69$) were prepared at UT Southwestern (Dallas TX) and sent to Baylor College of Medicine (David Wheeler, Houston TX) for paired-end RNA sequencing. The analysis was then performed at UT Southwestern: Reads were aligned to the human reference genome GRCh38 using STAR-2.7[97] (https://github.com/alexdobin/STAR) and FPKM values were generated with cufflinks-2.2.1[98] (http://cole-trapnell-lab.github.io/cufflinks/). All data were then pooled, upper-quartile normalized[99], and log-transformed.

**Cell culture**. All SCLC cell lines used in these studies were originally established in the John D. Minna and Adi F. Gazdar laboratories. The cultured SCLC cell lines were obtained from both the National Cancer Institute (NCI) and Hamon Cancer Center (HCC) libraries. Cells were cultured in RPMI-1640 media (Sigma Life Science, St. Louis, MO) supplemented with 5% Fetal Bovine Serum (FBS). RPMI-1640 supplemented with 5% FBS will be referred to as R5. All cells were incubated in NuAire (NuAire, Plymouth, MN) humidified incubators at 37 °C at 5% CO$_2$. All cell lines were regularly tested for mycoplasma contamination (Bulldog Bio, Portsmouth, NH) and fingerprinted using a PowerPlex 1.2 Kit (Promega, Madison, WI) to confirm the cell line identity.

**Establishing adherent H69 (H69-AD)**. The early passage of the parental H69 cell line grew as a mixture of floating and adherent cells. To enrich for adherent cells, the floating population of H69 was washed off during growth media replacement and fresh media was provided for the expansion of the remaining adherent cells. This was repeated until every passage grew as adherent cells with few to no suspension cells. This derived adherent subline was designated as H69-AD.

**Drug–response assay**. Cisplatin and Etoposide were obtained from Selleck Chemicals LLC, USA. 5000 cells of H69 and H69-AD were cultured in 100 μl R5 growth media per well in ultra-low adherent, clear, round bottom, 96 well plates (BD Biosciences, USA) for 48 h. An additional 100 μl R5 plus either a control (DMSO) or drug was added to the plate. 96 h after drug treatment, each cell line was assayed using the Cell-Titer-Glo reagent (Promega, Inc.). The fluorescence intensity was recorded at 570 nM. A standard 4-parameter log-logistic fit between the survival rate and the dosage was generated by the "drm" function from the R package "drc"[100].

**Xenograft models for parental and chemoresistant SCLC tumors**. Subcutaneous xenograft in NSG mice was derived from direct implantation of untreated H1436 cells or re-implantation of chemoresistant tumors after 4 cycles of Cisplatin and Etoposide (EC), or plus 4 cycles of Cisplatin (reduced from EC due to toxicity). Specifically, a million H1436 cells were resuspended in 100 μl mixture of serum-free RPMI-1640 and Matrigel (BD Bioscience #356237) at 1:1 ratio and immediately injected in the flank of 6–8-week-old female NSG mouse (Jackson Laboratory #005557). Mice were randomized after tumor cell injection. Treatment starts after a week when the tumor becomes palpable. 5 mg/kg/w Cisplatin (Sigma P4394) in saline, 10 mg/kg/w Etoposide (Sigma E1383) in 30% PEG 300 (Sigma 202371) were freshly prepared and administered by intraperitoneal injection, for 4 cycles in total to obtain the first group of chemoresistant tumors. An additional 4 cycles of Cisplatin were administered in the second group of mice with further potentiated chemoresistance. To harvest the tumor, 10 ml digestion media was used per mouse. This was prepared freshly by supplementing 9 ml HBSS with 1 ml type IV collagenase, 50 μl DNase II and 50 μl 1 M CaCl$_2$. Tumors were collected and placed in HBSS immediately following dissection. A fraction of the tumor was cut into a few pieces and flash-frozen in liquid nitrogen to be saved in aliquots for molecular assays. The remaining chunk was finely minced with a sterile scalpel blade. For re-implantation, the minced tissue was resuspended in digestion media, rotated at 37 °C for 20 min, filtered through a 40 μm filter, centrifuged at $300 \times g$ for 5 min.

**Quantitative reverse transcription PCR**. ~20 mg flash-frozen tumor fragments were weighed out and homogenized in 1 ml TRIzol (Invitrogen #15596-026) in Precellys tissue homogenizing mixed beads kit (Cayman Chemical #10409). 0.2 ml chloroform (Fisher #S25248) was added to the TRIzol lysate and the mixture was vortexed for 10 s and centrifuged at $12,000 \times g$ for 15 min at 4 °C for phase separation. 450 μl aqueous phase was collected, mixed well with 0.5 ml isopropanol (Fisher #A451-1), and precipitated RNA was collected by centrifugation at $12,000 \times g$ for 10 min at 4 °C, The RNA pellet was rinsed in 1 ml 75% ethanol, then dissolved in 100 μl deionized water by incubating at 55 °C for 5 min. 500 ng total RNA was reverse-transcribed to cDNA in a 20 μl reaction with 4 μl iScript reverse transcription supermix (Bio-Rad #1708841) at 25 °C for 5 min, 46 °C for 20 min, and 95 °C for 1 min. The mixture was then 1:5 diluted with deionized water. Target sequences in cDNA library were amplified in 10 μl qPCR reaction (5 μl SYBR Green supermix (Bio-Rad #1725121), 0.675 μl 2.5 μM primer mix and 0.45 μl diluted cDNA) at 95 °C 10 s, 60 °C 30 s, for 40 cycles. All procedures were performed under RNase-free conditions unless specified. For data analysis, the median was taken from triplicates, normalized by Ct values of control gene PPIA, exponentiated with base 2 then divided by the median of parental samples. Primer sequences are provided in Supplementary Table 1.

**Study approval**. The protocol of collecting human SCLC tumor tissue for research was approved by the Ethics Committee of Tianjin Medical University General Hospital. Written informed consent was received from participants prior to inclusion in the study. Specimen collection did not interfere with standard diagnostic and therapeutic procedures. All mouse procedures were performed with the approval of the University of Texas Southwestern Medical Center Institutional Animal Care and Use Committee.

**Statistics and reproducibility**. All statistical analyses were performed with R[101]. Pearson correlation was used to assess the association between continuous variables. t-test was used for group comparison. Pearson's chi-squared test was used to test for association between categorical variables. Statistical significance was set at $p \leq 0.05$. Benjamini–Hochberg procedures were used for generating p-values adjusted for multiple comparisons.

**Reporting summary**. Further information on research design is available in the Nature Research Reporting Summary linked to this article.

## Data availability

The RNA-seq gene expression data from UTSW SCLC has been added to dbGaP (accession phs001823.v1.p1)[102]. SCLC tumor microarray data used in this study have been deposited to GEO with accession id GSE149507. Source data used for the figures presented in this manuscript have been deposited in the Dryad Digital Repository (https://datadryad.org/stash/share/BkmPdMrwhae1VxhkkSLIG_532FLCqcYiMFUpY1yKmGA). All other data are presented in Supplementary Data 1–5 or available from the corresponding authors upon reasonable request.

## Code availability

Scripts and input data used for this manuscript have been deposited in the Dryad Digital Repository (https://datadryad.org/stash/share/BkmPdMrwhae1VxhkkSLIG_532FLCqcYiMFUpY1yKmGA).

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

## Acknowledgements

Support for this work comes from the National Institutes of Health [1R35GM136375, R35CA22044901, 5P30CA142543, 5U01CA213338-04, 3P50CA070907, and R01GM115473], and the Cancer Prevention Research Institute of Texas [RP180805]. We dedicate this work to Dr. Adi F. Gazdar, who had initially conceptualized and led this project until his passing in December 2018. We thank Ms. Jessie Norris for proofreading the manuscript.

## Author contributions

Conception and design, A.F.G. and L.C.; development of methodology, L.C., L.G., and T.W.; acquisition of data, L.C., H.L., F.H., J.F., L.G., J.C., Y.L., Y.-A.Z., D.D., V.S., K.P., C.S.K., C.Y., A.A., K.H., M.P.-Z., and B.D.; analysis and interpretation of data, L.C., J.F., G.J., L.Y., and W.Z.; writing, review and/or revision of the manuscript, L.C., H.L., E.A., R.J.D., and J.D.M.; study supervision, D.S.S., I.I.W., J.E.J., G.X., J.D.M., Y.X., and A.F.G.

## Competing interests

J.D.M. receives licensing fees from the NCI and UT Southwestern to distribute cell lines. R.J.D. is on the advisory board for Agios Pharmaceuticals. D.S.S. and W.Z. are currently employed by Genentech Inc. and own stock in Roche Holdings. I.I.W. is a speaker at Medscape, MSD, Genentech/Roche, PlatformQ Health, Pfizer, AstraZeneca, Merck; receives research support from Genentech, Oncoplex, HTG Molecular, DepArray, Merck, Bristol-Myers Squibb, Medimmune, Adaptive, Adaptimmune, EMD Serono, Pfizer, Takeda, Amgen, Karus, Johnson & Johnson, Bayer, Iovance, 4D, Novartis, and Akoya; and is on the advisory boards for Genentech/Roche, Bayer, Bristol-Myers Squibb, AstraZeneca/Medimmune, Pfizer, HTG Molecular, Asuragen, Merck, GlaxoSmithKline, Guardant Health, Oncocyte, and MSD. The remaining authors declare no competing interests.
