## [Peer Review File · Communications Biology]

REVIEWERS' COMMENTS:

Reviewer #1 (Remarks to the Author):

In the revised manuscript, the authors performed additional bioinformatic analyses of publicly available datasets to address criticisms and comments that were raised during the initial review. While the effort to analyze additional transcriptomic datasets is impressive and convincing, this study remains quite descriptive due to the lack of functional experiments that were requested by several reviewers. Specifically, the biological significance and mechanism behind the relationship between the expression of inflammatory modules and neuroendocrine signature remain unclear. Having said this, I am aware that many of the functional experiments requested are likely outside the scope of the current study.

While the manuscript remains descriptive, I do think it brings forward a new finding based on an impressive analysis of publicly available datasets. I would like to see that the authors more clearly acknowledge the limitations of this study and modified the results and discussion accordingly.

Reviewer #2 (Remarks to the Author):

In their revised manuscript, the authors have addressed all major concerns and have also enhanced the clarity in the presentation of their data. While the main weakness of the manuscript remains that there is a lack of functional experimental evidence that would bolster the author's conclusions, the analytical work is quite substantial and well-executed. In both the main manuscript and rebuttal comments, the authors have made intelligent use of existing published data and the consistency of their observations across multiple datasets helps to increase confidence in their correlative findings. Finally, the reported observations would be of great interest to others in the field and are likely to inspire further investigations to dissect the precise role of the neuroendocrine state in potentially dictating tumor-immune cell interactions.

Reviewer #3 (Remarks to the Author):

All the major comments are satisfactorily answered by the authors. However, the lack of experimental emphasis on understanding the mechanism remains to be a major deficiency. Some additional data included raises additional questions that though may be beyond the scope of the current manuscript. In overall, authors have satisfactorily answered the queries and made a lot of improvements to the manuscript with additional substantive content.

We thank the reviewers for taking time out of their busy schedules to go over our revised manuscript and read our rebuttal letter. We are very glad that reviewers 2 and 3 agreed that we have satisfactorily addressed all of their comments. We also recognize that all three reviewers have pointed out the lack of functional experiments with mechanistic insights is the major flaw of our paper, but we appreciate that they have also considered these as beyond the scope of this study. Echoing with their shared concern, we have acknowledged this limitation in the discussion section of our paper that “As a limitation of our current study, the biological significance and mechanism of ISG repression in healthy and cancer cells of NE lineage remains to be determined with additional functional studies”. We hope that with this modification, our manuscript will be acceptable for publication.

The reviewers’ comments verbatim are in blue italics. Our responses are in black text.

REVIEWER #1 – EXPERT IN LUNG CANCER GENOMICS (REMARKS TO THE AUTHOR):

“In the revised manuscript, the authors performed additional bioinformatic analyses of publicly available datasets to address criticisms and comments that were raised during the initial review. While the effort to analyze additional transcriptomic datasets is impressive and convincing, this study remains quite descriptive due to the lack of functional experiments that were requested by several reviewers. Specifically, the biological significance and mechanism behind the relationship between the expression of inflammatory modules and neuroendocrine signature remain unclear. Having said this, I am aware that many of the functional experiments requested are likely outside the scope of the current study.

While the manuscript remains descriptive, I do think it brings forward a new finding based on an impressive analysis of publicly available datasets. I would like to see that the authors more clearly acknowledge the limitations of this study and modified the results and discussion accordingly.”

Response:

We thank the reviewer for pointing out that our analysis effort is impressive and convincing. We also agree with the reviewer on the limitation of the study. We have now acknowledged this specifically in our discussion section that “As a limitation of our current study, the biological significance and mechanism of ISG repression in healthy and cancer cells of NE lineage remains to be determined with additional functional studies”. We do hope we will be able to find the resources to perform the experiments to find the answers, or inspire others to uncover the exact mechanism underlying these intriguing findings.

REVIEWER 2 - EXPERT IN BIOINFORMATICS AND RNA-SEQ (REMARKS TO THE AUTHOR):

“In their revised manuscript, the authors have addressed all major concerns and have also enhanced the clarity in the presentation of their data. While the main weakness of the

manuscript remains that there is a lack of functional experimental evidence that would bolster the author's conclusions, the analytical work is quite substantial and well-executed. In both the main manuscript and rebuttal comments, the authors have made intelligent use of existing published data and the consistency of their observations across multiple datasets helps to increase confidence in their correlative findings. Finally, the reported observations would be of great interest to others in the field and are likely to inspire further investigations to dissect the precise role of the neuroendocrine state in potentially dictating tumor-immune cell interactions.”

Response:

We thank the reviewer for appreciating our analytical work. As the lack of functional experimental evidence is a shared concern among three reviewers, we have now inserted into our discussion that “As a limitation of our current study, the biological significance and mechanism of ISG repression in healthy and cancer cells of NE lineage remains to be determined with additional functional studies”. We hope that as the reviewer has anticipated, our work will be an inspiration to the field for further investigation into this problem.

REVIEWER 3 - EXPERT IN IMMUNOGENOMICS (REMARKS TO THE AUTHOR):

“All the major comments are satisfactorily answered by the authors. However, the lack of experimental emphasis on understanding the mechanism remains to be a major deficiency. Some additional data included raises additional questions that though may be beyond the scope of the current manuscript. In overall, authors have satisfactorily answered the queries and made a lot of improvements to the manuscript with additional substantive content.”

Response:

We thank the reviewer for considering our answers satisfactory. Again, we have now pointed out the limitation of our work in the discussion section “As a limitation of our current study, the biological significance and mechanism of ISG repression in healthy and cancer cells of NE lineage remains to be determined with additional functional studies” to acknowledge this caveat. As the reviewer has pointed out that the additional data we included raises additional questions. We do hope we will be able to secure resources to work on these intriguing problems in the future.